# Understanding the impact of the SARS-COV-2 pandemic on hospitalized patients with substance use disorder

**Caroline King**[1,2]*, **Taylor Vega**[2], **Dana Button**[2], **Christina Nicolaidis**[1,3],
**Jessica Gregg**[1,4], **Honora Englander**[1]

**1** Department of Medicine, Oregon Health & Science University, Portland, OR, United States of America,
**2** School of Medicine, Oregon Health & Science University, Portland, OR, United States of America, **3** School
of Social Work, Portland State University, Portland, OR, United States of America, **4** De Paul Treatment
Centers, Portland, OR, United States of America

* kingca@ohsu.edu

of Medicine, UNITED STATES

**Data Availability Statement:** We are unable to
share data because data are potentially identifying
and contain sensitive patient information. These
restrictions are imposed by our ethics committee.

## Abstract

### Background

The SARS-COV-2 pandemic rapidly shifted dynamics around hospitalization for many communities. This study aimed to evaluate how the pandemic altered the experience of healthcare, acute illness, and care transitions among hospitalized patients with substance use disorder (SUD).

### Methods

We performed a qualitative study at an academic medical center in Portland, Oregon, in Spring 2020. We conducted semi-structured interviews, and conducted a thematic analysis, using an inductive approach, at a semantic level.

### Results

We enrolled 27 participants, and identified four main themes: 1) shuttered community resources threatened patients' basic survival adaptations; 2) changes in outpatient care increased reliance on hospitals as safety nets; 3) hospital policy changes made staying in the hospital harder than usual; and, 4) care transitions out of the hospital were highly uncertain.

### Discussion

Hospitalized adults with SUD were further marginalized during the SARS-COV-2 pandemic. Systems must address the needs of marginalized patients in future disruptive events.

## Introduction

The SARS-COV-2 pandemic has profoundly impacted communities around the globe. Marginalized communities, where people are excluded from full economic, sociopolitical, and cultural participation [1, 2], have disproportionately borne the harms of the pandemic [3–7].

Oregon Health & Science University's Institutional Review Board can be contacted by emailing irb@ohsu.edu, or emailing Kathryn Schuff, Program Manager, at schuffk@ohsu.edu.

**Funding:** CK and HE were supported by grants from the National Institutes of Health, National Institute on Drug Abuse (UG1DA015815/ R01DA037441). This publication was also made possible with support from the Oregon Clinical and Translational Research Institute (OCTRI), grant number UL1TR002369 from the National Center for Research Resources (NCRR), a component of the National Institutes of Health (NIH), and NIH Roadmap for Medical Research. CK was also supported by the National Center for Advancing Translational Sciences, National Institutes of Health, through Grant Award Number TL1TR002371. The content is solely the responsibility of the authors and does not necessarily represent the official views of the NIH. The funders had no role in study design, data collection and analysis, decision to publish, or preparation of the manuscript.

**Competing interests:** The authors have declared that no competing interests exist.

People with substance use disorder (SUD) experience marginalization in many social systems [8–10], including in healthcare [11–13]. Historically, this has manifested as healthcare systems that, at best, fail to adapt to function in the ways people with SUDs need to thrive [14], and at worst, actively drive participants from care through stigmatizing and discriminatory encounters [10].

Hospitalization may be a particularly challenging time for people with SUDs, as they commonly experience untreated withdrawal and pain and work to mitigate hospital-based social control (e.g. the rules and regulations that dictate acceptable behavior in hospitals) [15]. Negative and stigmatizing experiences during hospitalization can lead patients with SUDs to avoid presenting to the hospital until illnesses are dire, leave the hospital against medical advice, and trust healthcare providers less than patients without SUDs [15–18]. Avoiding healthcare systems, or leaving the hospital without completing recommended therapy, carries increased risk during a pandemic [19]. However, hospitals are also in a unique position to support patients with SUD, by providing evidence-based, compassionate care, including harm reduction tools, and by improving systems to meet the needs of people with SUD [20, 21].

The SARS-COV-2 pandemic rapidly and dramatically shifted care dynamics before, during, and after hospitalization for many communities. To help prevent the spread of SARS-COV-2 hospitals enacted many changes, including limiting visitors and reducing or eliminating in-person visits between providers and patients [22–24]. The SARS-COV-2 pandemic and mitigation strategies may have uniquely impacted people with acute illness and SUD. As we assess the ways in which hospitals' responses to the pandemic have succeeded and failed, we must include voices of people at the margins, and specifically, voices of people with SUD. By conducting qualitative research directly with people who are marginalized, we can understand how hospital policy changes have impacted them, which can help healthcare systems learn how to better care for patients in the future. Their experiences can highlight unique opportunities and gaps that can inform usual hospital care, as well as ongoing and future emergency responses. This study aimed to evaluate how the SARS-COV-2 pandemic has altered the experience of healthcare, acute illness, and care transitions among hospitalized patients with substance use disorder.

## Methods

### Setting and study design

We performed a qualitative study at an urban academic medical center in Portland, Oregon. This hospital is a 576-bed Level 1 Trauma Center located on the top of a hill in the city [25]. OHSU sees a wide variety of patients for acute medical and surgical illness, and often receives transfers of patients from smaller hospitals across Oregon. In this study, all participants saw an inpatient addiction medicine consult service called the Improving Addiction Care Team (IMPACT). IMPACT is the only inpatient addiction medicine consult service in the state of Oregon, though some other hospitals provide medication for Opioid Use Disorder in inpatient settings. IMPACT cares for adults with SUD, including but not limited to opioids, alcohol, and methamphetamines. The team includes addiction medicine providers, social workers, and peers with lived experienced in recovery. Earlier work describes IMPACT development, intervention, and outcomes [26–29].

### COVID-19 related changes and data collection timeline

OHSU implemented COVID-19 modified operations on March 23, 2020. Initially, the hospital restricted all patient visitors and encouraged clinicians who could to tele-work to preserve personal protective equipment (PPE) and support physical distancing [30]. For the first month of

modified operations, IMPACT medical providers predominantly consulted via telephone, reserving rare in-person visits for patients with cognitive issues, severe unmanaged withdrawal, or inability to participate in telephone visits. IMPACT social workers maintained some in-person hours but shifted primarily to telephone, and peers worked exclusively by phone. Despite best efforts, IMPACT could not perform patient visits via video, as most patients lacked smartphones and hospital rooms were not equipped with video technology. IMPACT leadership tried to secure personal electronic devices (e.g. smartphones, tablets) for patients to support video communication, but issues of cost and competing priorities were barriers and efforts did not come to fruition until after the study window. Researchers conducted individual telephone interviews between April 15 and May 29, 2020. Starting in mid-April of 2020, as hospital PPE was more available–and because of limitations of telephone-based care– medical providers transitioned back to in-person visits. In May, peer mentors resumed in-person visits. Social workers continued alternating telephone and in-person work due to the need for physical-distancing in office spaces. Overall, IMPACT returned to in-person operations sooner than many hospital services. Oregon and Portland have had low rates of cases of SARS-COV-2 and fatalities compared to states and cities nationally. As of late September 2020, Oregon reported 730 cases of COVID-19 per 100,000 people to the CDC, the second lowest of six rate-groups in the United States [31].

### Participants and data collection

Our institutional IRB waived the requirement for written consent given the inability to meet patients in person because of university-wide restrictions, and because of the minimal risk associated with our study. To recruit patients, we called the rooms of hospitalized patients with SUDs who were referred to IMPACT. We ascertained patient interest in learning about the study, and verbally consented patients to participate. Patients were offered $25 gift cards to thank them for participating.

For the quantitative survey, we included nine demographics questions and ten questions from the Brief Addiction Monitor [32]. The Brief Addiction Monitor has previously been used in inpatient settings with patients with substance use disorders [33, 34]. We also assessed participant readiness-to-change by substance, using four questions our team uses in research to assess substance use ("Please describe how you feel about your alcohol/opioid/amphetamine or methamphetamine/other drug use right now: 1) I do not want to think about cutting back or quitting, 2) I want to cut back but I am not ready to quit, 3) I want to quit, and 4) I quit before I was admitted to the hospital"). We developed the last question because we have been unable to find a measure that adequately measures readiness to change, by substance, for patients hospitalized with substance use disorders. We piloted tested the full survey among research team members prior to the start of the research project.

Semi-structured interviews explored participants' social networks; social services access; substance use, harm reduction, and treatment; housing; and experiences with medical care during the SARS-COV-2 pandemic (S1 File). Typically, interviews lasted 20–45 minutes.

The interview team consisted of three medical students, two female and one male, all interested in addiction medicine. Prior to the start of the study, two researchers who are experts in qualitative researcher (CN, HE) reviewed qualitative research steps for conducting interviews with the three medical students (CK, TV, DB), and oversaw the project for its duration. Then, one student with more qualitative experience (TV) conducted a mock interview with a student with less qualitative experience (DB) and gave feedback as to how to elicit rich qualitative data. Each of the three interviewers then did an initial one to two interviews. These interview recordings were transcribed, and the full research team reviewed transcripts and provided

feedback on interviewing techniques. Researchers securely recorded interviews via Webex and interviews were transcribed verbatim (Landmark Transcription Services).

### Data analysis

We conducted a thematic analysis, using an inductive approach, at a semantic level (e.g. where interview data were taken at face value) [35, 36]. The research team (CK, TV, DB, HE) reviewed five transcripts and generated initial codes. Three researchers (CK, TV, DB) then used an iterative approach to apply codes to all transcripts and update the codebook. All interviews were coded by at least two researchers, who met to regularly reconcile codes. The full team reviewed codes and identified preliminary themes, using a subjective heuristic for determining significance. A significant theme needed to: 1) be expressed by multiple participants; 2) be expressed as a central concern, 3) describe patient experiences related to acute illness or SUD, and the SARS-COV-2 pandemic. Researchers then re-reviewed primary data, chose representative quotations, and finalized themes. We used Dedoose [37] to manage qualitative data.

We used descriptive statistics to describe participant characteristics. We used Stata 16 [38] for quantitative analyses.

The Oregon Health & Science University Institutional Review Board approved this study (IRB #21399).

## Results

### Participant characteristics

We approached 42 patients via telephone to participate, and enrolled 27 participants. Of the 15 patients who declined to enroll, 8 said that they were not interested, 2 were managing acute illness and did not want to be on the phone, and 5 did not provide a reason for not participating. Of those who enrolled, most were male (59.2%), White (81.4%), and had a regular source of primary care (74.1%). Participants most commonly had opioid use disorder (OUD) (59.3%), methamphetamine use disorder (MethUD) (40.7%), or alcohol use disorder (AUD) (40.7%) (Table 1). Nearly three-quarters of participants (74.1%) stated that they did not have enough money to pay for basic necessities like food and clothing, and nearly half did not have a cell phone (48.1%).

Most participants reported that their substance use changed during the pandemic. In qualitative data, nearly everyone who used alcohol noted that their alcohol consumption increased during the pandemic. In contrast, participants who used methamphetamine described using less methamphetamine because of rapidly increasing costs and decreasing quality, while participants using heroin or other opioids noted use did not change or increased.

In qualitative analysis, we identified four themes and ten subthemes (in italics) related to acute illness, system stability and care transitions (Table 2). The shuttering of community resources threatened patients' basic survival adaptations before admissions. Additionally, changes in outpatient care increased reliance on hospitals as safety nets. Once hospitalized, hospital policy changes made it harder to stay, and care transitions out of the hospital were more uncertain than before the pandemic. We describe these below.

First, the **shuttering of community resources threatened patient's basic survival adaptations before admission.** During the SARS-COV-2 pandemic, participants reported *intensified insecurity around basic needs*, including accessing food, employment, housing, social services, and hygiene facilities. Each system that participants described was harder to access either because of increased demand (food), or because participants could not reach offices virtually, physical offices were closed, or both (social services, hygiene facilities).

**Table 1. Participant demographics.**

| Variable | Study participants N (%) or Avg (SD) |
|---|---|
| **Age (years)** | 48.9 (11.5) |
| **Gender (Male)** | 16 (59.2%) |
| **Race** | |
| White | 22 (81.4%) |
| Black | 1 (3.7%) |
| American Indian/Alaska Native | 5 (18.5%) |
| More than one race | 1 (3.7%) |
| Refused | 1 (3.7%) |
| **Hispanic** | 1 (3.7%) |
| **High-school education** | 23 (85.2%) |
| **Currently homeless** | 10 (37.0%) |
| **Have enough income to pay for necessities in past 30 days** | 7 (25.9%) |
| **Have a cell phone** | 14 (51.9%) |
| **Usual place of primary care** | 20 (74.1%) |
| **SUD Diagnoses** | |
| Opioid Use Disorder | 16 (59.3%) |
| Methamphetamine Use Disorder | 11 (40.7%) |
| Cocaine Use Disorder | 3 (11.1%) |
| Alcohol Use Disorder | 11 (40.7%) |
| Benzodiazepine Use Disorder | 2 (7.4%) |
| Other Use Disorder | 4 (14.8%) |
| **Attended self-help meeting in past 30 days** | 2 (7.4%) |
| **Bothered by cravings in past 30 days (n = 26)** | |
| Not at all | 10 (38.5%) |
| Slightly | 1 (3.8%) |
| Moderately | 2 (7.7%) |
| Considerably | 0 |
| Extremely | 13 (50.0%) |

For participants without housing, the pandemic *heightened the toll of homelessness*. Some participants perceived increased stigma associated with homelessness during the pandemic, noting that others were less willing to engage with them for fear of contracting SARS-COV-2. Restaurants and stores were closed, eliminating options to warm-up, sleep, access Wi-Fi, or panhandle outside. One participant, who was homeless and admitted with heart failure, shared that he and his wife struggled now that the physical McDonald's location was closed:

"We'd get up in the morning and go over to McDonald's and get the dollar coffee and sit there, and surf the web on your phone. It was all it was. Following the COVID-19—what was gonna happen next? Now they're gonna close this, they're gonna close that, oh, crap. Now you can't do anything except through the drive-through. We just watched it tumble down." (Participant 14).

Some participants became newly homeless or housing-unstable because of arguments and strained relationships in the context of the pandemic. Some participants left romantic partners, family or friends to live outside, while others moved between different homes. Participants who shifted to living outside described having nowhere else to turn during the pandemic.

**Table 2. Qualitative themes and subthemes.**

| Theme | Subtheme |
|---|---|
| **Shuttering of community resources threatened patient's basic survival adaptations before admission** | Intensified insecurity around basic needs |
| | Heightened toll of homelessness |
| | Virtual meetings limited engagement and access to outpatient substance use services |
| **Changes in outpatient care increased reliance on hospitals as safety nets** | Lack of technology severely limited access to the outpatient care that remained open |
| | Participants turned to the hospital when they were sick, particularly if they did not own a phone |
| **Changes in hospital policy made receiving care and staying in the hospital harder than usual.** | Hospitalization was more isolating than ever |
| | PPE solidified the gravity of the pandemic and signaled strict rule-enforcement |
| | Inpatient addiction care was essential during the pandemic |
| **Transitions out of hospital care were highly uncertain** | Housing was a top worry post-discharge |
| | Participants worried about new and ongoing vulnerabilities following acute illness |

Next, a shift to *virtual meetings limited engagement and access to outpatient substance use services*. Many support group meetings (e.g. Narcotics Anonymous) moved online or stopped altogether during the pandemic. This led some participants to return to substance use, and others to increase use. The shift online was particularly harmful to people who lacked smartphones or internet. One participant, who predominately used opioids and benzodiazepines and did not own a phone, described being "depressed" and noted that an inability to access recovery meetings "made my use go up because I felt lost." (Participant 17). Importantly, participants reported how syringe distributions sites responded differently to the pandemic: they largely remained open, they promoted physical distancing, and offered more supplies to reduce the number of trips people had to make. Participants found harm reduction services to be an "actually helpful" system in unstable times.

Some patients felt that these acute changes in society, and the subsequent changes in their substance use, contributed, or led, to their hospitalization. For others, personal struggles to survive during the pandemic added additional layers of complexity to hospitalization and post-discharge plans.

Second, we found that **changes in outpatient care increased reliance on hospitals as safety nets**. Just as not having a phone excluded patients from virtual recovery meetings, it also *severely limited access to the outpatient care that remained open*. Without a phone or other electronic device, participants reported that they could not contact outpatient providers to make appointments or seek guidance. Participants described that societal shifts to communicating via technology "threw a wrench in the works" during the pandemic. Some participants blamed outpatient care facilities that were not allowing in-person visits. One participant with no phone described,

> "I've had hellacious problems. . .. Before I came to the hospital, I couldn't get a doctor's appointment for the life of me to get that port out of my chest. . .I'm pretty sure that's what put the clot in my heart, you know?" (Participant 17).

Another participant expressed worry that shuttered outpatient care had delayed his access to a diagnostic colonoscopy and made him sicker, eventually landing him in the hospital.

"They bumped [the colonoscopy] and said they'd get a hold of me when they were doing them again. Then, they finally got a hold of me, and by then I was really sick. . .. Just I hope it's not too late." (Participant 24).

While not having a phone was a unique stress for participants trying to access outpatient services, even participants with phones described challenges accessing care. Some participants who contacted their primary care office for a medical issue felt that in pre-COVID times, their primary care provider would have seen them in-office, but because of the pandemic and because they were very sick, they were sent to the emergency department instead.

When outpatient care was open only for virtual visits, *participants turned to the hospital when they were sick*, *particularly if they did not own a phone*. Participants saw this as a last resort, and described worries of contracting SARS-COV-2 and being separated from family and friends during such a vulnerable time of acute illness. One participant, admitted with MRSA bacteremia and cellulitis, described presenting to the hospital emergency department, when her friend and partner were unable to accompany her:

"I was afraid to come in at all and really wanted and needed them with me, that support. . ..I made it as far as the outside doors to the emergency department. They had those COVID like tents set up. . . I just burst into tears and turned around. . .. I just kept walking and went to a bus stop and went home. I was afraid to go in and I didn't. . ..I chickened out the first time and was afraid to go in, so not being able to have somebody with me really affected me. I had to work up the courage to do it all by myself." (Participant 19).

Participants saw hospitalization as a last, essential resort, and often tried to avoid it by attempting to seek care elsewhere or delaying their presentation to care. Because care elsewhere was often inaccessible, particularly to participants without phones or internet, there was increased reliance on hospitals as safety nets during the early months of the SARS-COV-2 pandemic.

**Changes in hospital policy made receiving care and staying in the hospital harder than usual.** For most participants, care teams would call patient rooms instead of visiting in person, if possible. While a few participants found this helpful, most felt it was difficult, and that receiving constant phone calls to their room was exhausting and made building trust with providers difficult. As one participant, who primarily used methamphetamine and opioids, shared,

"I much prefer [seeing providers] in-person. The over-the-phone. . .it's a little bit more cold and impersonal. . .you've never met before, you've never shook hands, you've never laid eyes on each other. . . There's something about meeting a person, and looking them in the eyes, and shaking their hand that I realize for me really means a lot. I can't just open up to somebody that I just talked to on the phone in the same way." (Participant 16).

Additionally, participants who had previously been hospitalized noted that *hospitalization was more isolating than ever* before for two reasons. First, participants had no in-person family and social support, due to restricted visitor polices aimed at preventing the spread of SARS-COV-2. Second the transition to telemedicine resulted in decreased interactions with healthcare providers, compounding widespread feelings of "isolation" in the hospital. One participant with alcohol use disorder shared,

"It's lonely. . .I have my cellphone, but I can't have visitors. There's hardly any nurses and doctors in the hallways. You really don't see anybody. Everybody's locked away." (Participant 23)

Simultaneously, when providers did enter participant rooms, they donned PPE. Participants found PPE to be both comforting, in that they expected providers to be wearing it, and anxiety-inducing, because watching providers don *PPE solidified the gravity of the pandemic and signaled strict rule-enforcement*. One participant, with opioid use disorder, noted:

"They're very strict on their procedures that they put in place for the virus. . . I don't blame anybody for that. They're trying to do the right thing. God forbid. I just hope this whole thing is over with sometime soon." (Participant 27).

Participants noted that *inpatient addiction care was essential during the pandemic*, and increasingly so as some participants newly returned to substance use. One participant shared that he felt IMPACT was

". . . very essential at this point in time, because I feel that I'm not the only person that's turning to drugs and alcohol. . .. I like the fact that [IMPACT] are there. I can call them. They'll come see me. They can talk to me about anything. I can talk to them about anything." (Participant 16).

As the pandemic progressed, IMPACT shifted from telemedicine to mostly in-person visits. For some participants, phone visits with IMPACT did not work. As one participant who was scared to present to the hospital because of SARS-COV-2 risk described, ". . .they were just voices on the phone. I [couldn't] even keep them straight, who was who, the name with the—I didn't know from one call to the next really." (Participant 19). Others noted how much they appreciated the in-person visits from IMPACT, describing ". . . I appreciate[d] the fact that you're not scared of the coronavirus to actually come and see me in my hospital room." (Participant 16).

Participants described inpatient addiction care as essential during hospitalization during the SARS-COV-2 pandemic, and highlighted the importance of in-person communication to build trust with addiction providers.

Finally, for study participants, **transitions out of hospital care were highly uncertain**. For nearly every participant, *housing was a top worry post-discharge. Participants worried about new and ongoing vulnerabilities following acute illness*. One participant with opioid use disorder, who lost his housing just before admission and was admitted for infection and frostbite, described:

"Even when I get out of here, I can't go back out on the streets and be like the status that I'm in right now. It just won't work . . .I'll be an open target. . . . They'll come take all your medications from you. They'll steal everything you own. . .. You got to fall asleep sometime." (Participant 27).

SARS-COV-2 heightened risks of these already vulnerable transitions, introducing uncertainty into where people could go, who they would be around, how they would limit risks for SARS-COV-2 infection, and because of the challenges they would face meeting basic survival needs. One participant noted,

"My biggest concern when I leave the hospital is relocating to a place where I know really not many people and where I can pick and choose the crowd that I allow myself to be surrounded by. My biggest concern is finding some type of safe, secure place that I could lay my head because I'm gonna have a lot of open tubes." (Participant 16).

Many participants described added uncertainty of how they would manage their health conditions, and other unknown post-discharge dynamics. Participants shared the experience of having few answers and many fears about the post-discharge period.

## Discussion

Our study describes the ways in which SARS-COV-2 has impacted the lives of hospitalized patients with SUD. Most participants reported increased marginalization because of policy changes. They reported the shuttering of community resources threatened their survival strategies, intensifying insecurity around basic needs and amplifying the toll of being homeless. Changes in outpatient care increased reliance on hospitals as safety nets. Having a phone and internet were essential to accessing outpatient care virtually, and in the absence of these, participants reported turning to the hospital when they were sick. Once admitted, changes in hospital policy made staying in the hospital harder than usual. Participants found the hospital isolating and had difficultly connecting with providers by phone. Participants described addiction care as essential during this time. Finally, discharge care transitions were even more uncertain than before the pandemic. Housing was a top concern, and participants worried about worsening challenges facing recovery from acute illness following discharge.

Our results map onto the large body of work that describes people who are marginalized, including people with addiction, as rarely considered before disaster strikes, and thus disproportionately impacted by disruptive events like pandemics and natural disasters [39–45]. Most research around addiction in the face of disruptive events has focused on challenges in rapidly shifting treatment systems (e.g. reintegrating those fleeing Hurricane Katrina into methadone clinics in Texas [45]; shifting buprenorphine prescribing to a different clinic following hospital flooding in New York [40]), in which treatment systems must quickly adapt to minimize interruptions in medication-based treatment. Participants in our study did not describe challenges accessing buprenorphine or methadone, but did describe challenges accessing addiction care like outpatient recovery meetings. Published studies also highlight increased substance use among people already using substances after disruptive events [41]. Participants in our study endorsed this trend, highlighting that given increased isolation and anxiety during the pandemic, substance use often increased for participants using alcohol and, sometimes, opioids.

However, outside of the context of treatment interruptions, little research has described the impact of disruptive events on the lives of people who use drugs. Participants in our study described that policy changes created threats to survival (increased homelessness, decreased access to outpatient care) before hospitalization with acute illness. Participants recognized they were returning to these same dynamics after discharge, and often wondered how they would manage illness in the face of compounding vulnerabilities. Care transitions, a vulnerable time for patients with substance use disorder in non-pandemic times [46, 47], were fraught with heightened uncertainty and worry. Participant experiences of system-failures highlighted that decision-makers within healthcare and government were unprepared to rapidly adjust to the needs of patients with SUD, particularly for patients who were also homeless.

As marginalization is driven by the socio-political and economic contexts of society [48], it is unsurprising that systems that permit marginalization were not prepared to respond to the SARS-COV-2 pandemic in the ways necessary to protect patients with addiction from harm. Importantly, however, two systems *were* prepared to mitigate suffering for patients with addiction. First, outside of the hospital, needle exchange services rapidly adapted to allow people to both decrease the risk of SARS-COV-2 transmission and access safer-injection supplies. Second, within the hospital, the addiction consult service continued to see patients with addiction and worked with study participants to mitigate heightened uncertainty and isolation in the

face of the pandemic, particularly around transitions out of the hospital. Both findings build on existing literature that show addiction consult services, in non-pandemic times, help build trusting patient-provider relationships [17], improve engagement in substance use treatment after hospital discharge [26], and help patients determine and pursue their healthcare goals [18, 27]. Participants described addiction consult services as especially important during the pandemic. Both addiction consult services and needle exchange services incorporate voices of people with lived experience with SUD from the earliest stages of planning [29, 49]. This natural programmatic inclination to seek and incorporate the needs of people with SUDs in non-pandemic times may have facilitated rapid responsiveness to the needs and goals of people with SUD during the SARS-COV-2 pandemic.

Our study has several limitations. First, we conducted our study at a single, academic medical center, which may limit generalizability, as community-based hospitals may see different challenges among patients with substance use disorders admitted to the hospital. Second, Oregon had relatively low rates of SARS-COV-2 infections during the study period, but this has since changed [31]. It is possible that patient experiences and priorities differ depending on community prevalence of SARS-COV-2; however, the strains imposed by shuttered services, hospital policies, and high-risk transitions would likely only be further compounded in settings with high SARS-COV-2 cases (for example, in Philadelphia, where treatment clinics did close [50]). Third, our sample had low racial and ethnic diversity. Because Black, Indigenous, and People of Color disproportionately suffered from death and other complications of SARS-COV-2 [51] and also disproportionately experience opioid-related harms, including during the pandemic [52], future work should explore intersecting vulnerabilities of SUD, hospitalization, and acute crises like pandemics in this population. Fourth, it is possible that our research does not encompass all challenges faced by marginalized patients in healthcare systems. While we focus specifically on patients with substance use disorders and highlight that addressing challenges for these patients may benefit other marginalized groups, there may be other, unique, challenges that should be explored through further research with other marginalized patient populations. Fifth, as is common in emergencies, policy within the hospital related to COVID-19 rapidly changed over the study period. We do not know to what degree these rapid changes may have impacted study results. Sixth, additional research is needed to understand how the SARS-COV-2 pandemic directly impacted substance use access, practices, and use. While we found differences in use practices and access by primary substance, this question is better answered using quantitative data.

Our findings have important implication for hospital providers, health systems, and policy makers. Hospital providers may need to reconsider what constitutes readiness for discharge during future disruptive events, and consider factors like access to technology, shelter, and ambulatory care–all of which were disrupted for participants in our study. Further, it is important for hospital providers to understand patient's intensified feelings of isolation, and the challenges to building trusting relationships with structural barriers, including virtual visits and PPE. To address this, providers may empathize with patient's circumstances, offer extra compassion or small gestures to connect with patients, and inquire as to how they can support patients during stressful times. Chua et al. [23] have derived guidelines for serious illness conversations held virtually in palliative care settings during the pandemic; aspects of these guidelines may be useful for working with any vulnerable population, and include acquainting the patient to the technology and responding to patient emotion.

In the face of outpatient closures, increasing disease and injury, or both, healthcare systems serve as a safety net for those with no place else to turn. Healthcare systems must be prepared for this role, not only for the general population, but also for patients who experience high degrees of marginalization, including patients with addiction. To do this, healthcare systems

must incorporate the voices of marginalized people into disaster planning. In our study, nearly half of participants did not own a phone. Healthcare system shifts to telemedicine and virtual recovery supports were suddenly inaccessible to patients in need. Globally, shifts to telemedicine have impacted not only patients with substance use disorders, but also patients who are older, with disabilities, with lower socioeconomic status, and who are homeless [53]. First steps towards more equitable systems should identify basic barriers to health that could be relevant in most disasters, like technology access, access to shelter, clean water and food, and essential medicines, and work to address these challenges to support a basic level of health and wellness in communities. Mitigating these challenges in non-pandemic times can ease the burden of potential patient surges during crises, but marginalized communities must be involved in these decisions. Addiction consult services demonstrated the importance of this, as patients highlighted that addiction consult services helped meet their needs during their inpatient stay, particularly during the pandemic. As hospitals work to incorporate patient voices into policy planning, addiction consult services may serve as a cultural broker to help immediately care for vulnerable patients (particularly as they transition out of the hospital) and communicate important needs with hospital leadership, from patient perspectives.

Finally, as policy-makers prepare for future pandemics, natural disasters, or other disruptions, they must consider effects on marginalized populations, and specifically people with SUD, as a priority. To prevent future potential harms, policy-makers should help alleviate the need to simply survive in non-pandemic times and help support access to housing, harm reduction and treatment services. To do this, policy-makers must include marginalized people at the decision-making tables. In preparation for future events, hospital and other leaders must develop partnerships with organizations that understand community needs of marginalized people, including those with SUD, to anticipate and plan for diverse patient needs during times of crisis.

## Conclusion

Hospitalized adults with SUD were further marginalized during the SARS-COV-2 pandemic. Hospitals providers and leaders play critical roles in creating systems that alleviate suffering and support people with SUD. To do this more effectively and in preparation for future disruptive events, healthcare systems must incorporate and amplify the voices of marginalized patients with SUD to alleviate suffering at the bedside and support safe transitions out of care. Responsive systems will be necessary to care for not only the masses, but the most marginalized, patients in the face of future disruptive events.

## Supporting information

**S1 File. Participant survey.** Survey for participants with substance use disorder, hospitalized during the SARS-COV-2 pandemic.
(PDF)

**S1 Poster.**
(PPTX)

## Acknowledgments

### Prior presentations

Preliminary results from this work were presented as an abstract at AMERSA in Fall 2020.

## Author Contributions

**Conceptualization:** Caroline King, Taylor Vega, Dana Button, Honora Englander.

**Data curation:** Caroline King, Taylor Vega, Dana Button.

**Formal analysis:** Caroline King, Taylor Vega, Dana Button, Christina Nicolaidis, Honora Englander.

**Investigation:** Caroline King, Honora Englander.

**Methodology:** Caroline King, Taylor Vega, Christina Nicolaidis, Jessica Gregg, Honora Englander.

**Project administration:** Caroline King, Honora Englander.

**Resources:** Honora Englander.

**Supervision:** Caroline King, Christina Nicolaidis, Honora Englander.

**Writing – original draft:** Caroline King.

**Writing – review & editing:** Caroline King, Taylor Vega, Dana Button, Christina Nicolaidis, Jessica Gregg, Honora Englander.

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
