## [Decision Letter · Decision Letter 0]

31 Dec 2020

PONE-D-20-30950

Understanding the impact of the SARS-COV-2 pandemic on hospitalized patients with substance use disorder

PLOS ONE

Dear Dr. King,

Thank you for submitting your manuscript to PLOS ONE. After careful consideration, we feel that it has merit but does not fully meet PLOS ONE’s publication criteria as it currently stands. Therefore, we invite you to submit a revised version of the manuscript that addresses the points raised during the review process.

Required changes:

1.  Deeper discussion in the introduction related to the impact of the pandemic and mitigation strategies on PSUD.

2. More detail in the Methods about the hospital population, representativeness of study participants, recruitment process and reasons for non-participation.

3. Reviewer 2 requests several definitions/clarifications of terms.

4.  Strengthening the literature review particularly surrounding issues of phone/internet access among this population and the challenges of utilizing telehealth.

5.  Attention to formatting of references. 

Recommended:

1.  Expanding the discussion of limitations and address the limitation of lack of a telephone may have had on recruiting patients.

2.  Reviewer 1 asks about how the timing of the pandemic may have influenced or shifted themes.  It may be challenging to disentangle timing of interviews with timing of COVID-related changes however more discussion of how the pace of COVID-related changes on data collection and patient perspectives is warranted.

We look forward to receiving your revised manuscript.

Kind regards,

Cynthia Sieck

Academic Editor

PLOS ONE

Additional Editor Comments:

This manuscript examines the experiences of people with substance use disorder during the COVID-19 pandemic and is a very timely and important issue to understand. The reviewer comments suggest important revisions which would improve the manuscript. Particularly:

1. More detail is needed in the methods section regarding the hospital population and how representative study participants are.

2. More detail is needed about recruitment as discussed by the reviewers.

3. A deeper discussion of how the pandemic itself as well as mitigation strategies related to the pandemic can impact the lives of PSUD would be beneficial. This would provide better context for the challenges this population faces.

Journal Requirements:

2. In the Methods, please discuss whether and how the questionnaire was validated and pre-tested. If these did not occur, please provide the rationale for not doing so.

"IMPACT is funded by Oregon Health & Science University and CareOregon.

CK and HE were supported by grants from the National Institutes of Health, National Institute

on Drug Abuse (UG1DA015815/ R01DA037441). This publication was also made possible with

support from the Oregon Clinical and Translational Research Institute (OCTRI), grant number

UL1TR002369 from the National Center for Research Resources (NCRR), a component of the

National Institutes of Health (NIH), and NIH Roadmap for Medical Research."

"CK and HE were supported by grants from the National Institutes of Health, National Institute on Drug Abuse (UG1DA015815/ R01DA037441). This publication was also made possible with support from the Oregon Clinical and Translational Research Institute (OCTRI), grant number UL1TR002369 from the National Center for Research Resources (NCRR), a component of the National Institutes of Health (NIH), and NIH Roadmap for Medical Research. The funders had no role in study design, data collection and analysis, decision to publish, or preparation of the manuscript."

4. We noted in your submission details that a portion of your manuscript may have been presented or published elsewhere.

"Preliminary results from this work were published virtually at AMERSA in Fall 2020. "

Please clarify whether this publication was peer-reviewed and formally published. If this work was previously peer-reviewed and published, in the cover letter please provide the reason that this work does not constitute dual publication and should be included in the current manuscript.

Reviewers' comments:

Reviewer's Responses to Questions

**Comments to the Author**

1. Is the manuscript technically sound, and do the data support the conclusions?

Reviewer #1: Yes

Reviewer #2: Yes

2. Has the statistical analysis been performed appropriately and rigorously? 

Reviewer #1: N/A

Reviewer #2: Yes

3. Have the authors made all data underlying the findings in their manuscript fully available?

Reviewer #1: No

Reviewer #2: Yes

4. Is the manuscript presented in an intelligible fashion and written in standard English?

Reviewer #1: Yes

Reviewer #2: Yes

5. Review Comments to the Author

Reviewer #1: This manuscript describes findings from a prospective qualitative study of 27 individuals with substance use disorder who were hospitalized during the COVID pandemic in an academic medical center in Portland, Oregon. What is challenging about the manuscript is that because only hospitalized patients with substance use disorder (PSUD) were interviewed, it is difficult to determine the extent to which the COVID-19 impacts describe in the paper are unique to this population or whether other patients in the hospital faced similar challenges. If this facility is a safety net hospital, many of the challenges around housing and other determinants of health in the context of the pandemic may not be all that different for other populations of patients. This limitation is not discussed in the paper.

Introduction: The Introduction clearly and succinctly lays out the key issues that people with substance use disorders (PSUD) face in engaging with the healthcare system. The authors also note the rapid shifts in protocols within hospitals in response to the SARS-COV-2 pandemic. Attention to two issues in the Introduction would strengthen this section. First, the authors contend (page 3), “The SARS-COV-2 pandemic and mitigation strategies may have uniquely impacted people with acute illness and SUD.” This may indeed be the case but 1-2 more sentences about these unique impacts would orient the reader to the authors’ line of reasoning. At the same time, the authors cannot actually provide evidence to support this claim because the study only includes hospitalized PSUD. Second, the statement, “As we assess the ways in which hospitals’ responses to the pandemic have succeeded and failed, we must include voices of people at the margins, and specifically, voices of people with SUD,” should also be explained further. For readers who may be less well-versed in, for example, community-engaged research, further articulating what can be gained by examining the perspectives of PSUD in designing healthcare systems and processes would strengthen the rationale for the research.

Methods, Setting and Study Design: If the authors provided 1-2 more sentences about the broader hospital and the patients typically served, it would better contextualize the findings.

In addition, it was somewhat difficult to tie together the state of SARS-COV-2 changes in the hospital during the specific timeframe of data collection. It appears that, to some degree, processes were stabilizing during the timeframe, but if the paragraph about SARS-COV-2 was woven into the description of when data were collected, it would be more clear.

Also were the measures of accessing healthcare from an existing measure or developed specifically for this study?

Methods, Participants and Data Collection: Adding one sentence that describes the training of the interviewing team is also warranted. Also, consider moving the information about the sample size to this paragraph rather than including as the first sentence in the Results section.

Results: The main findings are clearly conveyed, with themes supported by representative quotations from the interview transcripts. The findings about the challenges for individuals without cell phones are a major theme, but this reviewer wondered how individuals with phones perceived the shift to telehealth. In several sections of the Results, more explicit comparisons of themes for those with (51% of the sample) and those without phones (49%) would be beneficial, as it is difficult to gauge the perspectives of the half of the sample who did have phones.

Discussion: The Discussion does a nice job linking the findings to some of the literature on natural disasters and pandemics. Some of the tone about policy-makers is a bit beyond scope of the data (e.g., “Within Oregon, local, state, and national policy-makers implemented strategies meant to curb the spread of SARS-COV-2 for the masses, but instead inadvertently threatened the survival of people living at the margins”).

A second consideration is if or how themes perhaps shifted over the time period of data collection. Over the 6 weeks of data collection, it seems that the hospital restrictions loosened to some degree (based on the description in Methods about the impact of the pandemic within the hospital), but this is not discussed in terms of the findings.

The Limitations are a bit perfunctory regarding the limitation of focusing on an academic medical center. An additional sentence that elucidates why that is a limitation would be beneficial. Also, given a rapid escalation in cases this fall in Oregon, perhaps a sentence about how timing may affect the findings is warranted.

The discussion about marginalized populations and telemedicine (page 15) would benefit from additional citations, as there have been concerns raised about unequal access to telemedicine both for SUD specifically but also for other health conditions.

References: The Reference list needs careful editing. There is variability in article titles in how the titles are capitalized, some author names seem erroneous (especially #24 and #37), there are inconsistencies in whether journal names are abbreviated or not, and some references pre-2020 lack volume and page numbers.

Reviewer #2: Thank you for inviting me to review this interesting and important paper. It presents the results from a qualitative study conducted with 27 patients hospitalized with substance use disorder during the initial phase of the COVID-19 pandemic in Spring 2020. The objective is clearly stated and appears to have been met (“evaluate how the SARSCOV-2 pandemic altered the experience of healthcare, acute illness, and care transitions among hospitalized patients with substance use disorder.”) The paper is well written and the findings are useful for informing improvements in clinical care for patients with SUD in pandemic settings and beyond. I especially appreciate the careful analysis, inclusion of patients’ voices, and discussion of important implications. I have one overall comment about the literature review and then some minor suggestions about ways that this paper could potentially be improved:

The literature review feels a bit thin in places. Although COVID-19 is newly-emerged infectious disease, there may be experiences from other disasters that could be cited in the Introduction (some is cited in the Discussion section but that feels a bit late – this context would be helpful up front). There are also new studies and commentaries emerging that make me wonder if the lit review could be updated a bit? For example, PMC7419278 discusses some innovations implemented in one clinical setting during the pandemic with implications for care of patients with SUD. Similarly there are a few specific topics (e.g., stigma in the Introduction) where additional recent qualitative studies could be cited (PMID: 30884432). Taken together, these studies underscore the implications of the findings of the present study that if care is not improved for patients with SUD during pandemics (in part by reducing stigma and mistreatment), individuals may leave the hospital against medical advice, which has implications for their own health and others’ (esp. given the infectious nature of COVID). So strengthening the lit review throughout would help.

Minor comments:

Introduction:

Topics like “hospital-based social control” warrant some defining and explaining (as this, in particular, is a relevant issue for this population). Take care throughout to avoid the use of jargon or other terms that a general audience of this journal may not be familiar with.

Methods:

It is unclear what a “prospective qualitative study” is. (also in the Abstract). If participants were enrolled sequentially that is fine but does make this “prospective” in the sense that it was a cohort or other type of study looking forward. And it does not sound like patients were interviewed more than once.

Please explain why there weren’t more efforts to implement telemedicine for this population initially. Experiences from other settings have shown that hospitals can provide wifi enabled devices to inpatients for consults (and even extending to after discharge).

It sounds like data collection was collected remotely but that is not specifically explained. Nor are the recruitment or consent/enrollment processes if this was all remote/virtual. Please explain these procedures and comment on any related challenges or facilitators as this could be informative to the research community.

Similarly, please include more details on IRB review and consent given that there was an unprecedented shift in research operations and oversight during this time period.

What does “at a semantic level” mean? This should be defined in the text.

Results: what were the reasons for declining participation or ineligibility?

Discussion: see earlier comment about moving some of the literature on disasters to the Introduction. This is optional but I feel would help to set the stage earlier on.

Limitations: there is increasing evidence that Black and Hispanic individuals are at higher risk of experiencing opioid-related harms, not just COVID.

6. PLOS authors have the option to publish the peer review history of their article (what does this mean?). If published, this will include your full peer review and any attached files.

Reviewer #1: **Yes: **Hannah Knudsen

Reviewer #2: No

---

## [Author Response · Author response to Decision Letter 0]

29 Jan 2021

Thank you. We have updated formatting to PLoS One Style throughout.

2. In the Methods, please discuss whether and how the questionnaire was validated and pre-tested. If these did not occur, please provide the rationale for not doing so.

For the quantitative survey, we included 9 demographics questions (assessing gender, race, ethnicity, housing status, income, education, zip code, cell-phone ownership, and marital status) and 10 questions from the Brief Addiction Monitor. The Brief Addiction Monitor has previously been used in inpatient settings with patients with substance use disorders. Finally, we asked patients about their readiness-to-change by substance, using four questions our team uses to assess substance use (Question: Please describe how you feel about your alcohol/opioid/amphetamine or methamphetamine/other drug use right now: 1) I do not want to think about cutting back or quitting, 2) I want to cut back but I am not ready to quit, 3) I want to quit, and 4) I quit before I was admitted to the hospital). We developed the last question because we have been unable to find a measure that adequately measures readiness to change, by substance, for patients hospitalized with substance use disorders. We piloted tested the full survey among research team members prior to the start of the research project. We have updated our text to include the following:

“For the quantitative survey, we included nine demographics questions and ten questions from the Brief Addiction Monitor (32). The Brief Addiction Monitor has previously been used in inpatient settings with patients with substance use disorders (33, 34). We also assessed participant readiness-to-change by substance, using four questions our team uses in research to assess substance use (“Please describe how you feel about your alcohol/opioid/amphetamine or methamphetamine/other drug use right now: 1) I do not want to think about cutting back or quitting, 2) I want to cut back but I am not ready to quit, 3) I want to quit, and 4) I quit before I was admitted to the hospital”). We developed the last question because we have been unable to find a measure that adequately measures readiness to change, by substance, for patients hospitalized with substance use disorders. We piloted tested the full survey among research team members prior to the start of the research project.” (Page 5, paragraph 3)

"IMPACT is funded by Oregon Health & Science University and CareOregon.

CK and HE were supported by grants from the National Institutes of Health, National Institute

on Drug Abuse (UG1DA015815/ R01DA037441). This publication was also made possible with

support from the Oregon Clinical and Translational Research Institute (OCTRI), grant number

UL1TR002369 from the National Center for Research Resources (NCRR), a component of the

National Institutes of Health (NIH), and NIH Roadmap for Medical Research."

"CK and HE were supported by grants from the National Institutes of Health, National Institute on Drug Abuse (UG1DA015815/ R01DA037441). This publication was also made possible with support from the Oregon Clinical and Translational Research Institute (OCTRI), grant number UL1TR002369 from the National Center for Research Resources (NCRR), a component of the National Institutes of Health (NIH), and NIH Roadmap for Medical Research. The funders had no role in study design, data collection and analysis, decision to publish, or preparation of the manuscript." Please include your amended statements within your cover letter; we will change the online submission form on your behalf.

Thank you. We have removed funding information from the manuscript, and have included our finalized funding statement in our cover letter.

4. We noted in your submission details that a portion of your manuscript may have been presented or published elsewhere. "Preliminary results from this work were published virtually at AMERSA in Fall 2020." Please clarify whether this publication was peer-reviewed and formally published. If this work was previously peer-reviewed and published, in the cover letter please provide the reason that this work does not constitute dual publication and should be included in the current manuscript.

We apologize for the confusing language- preliminary results from this work were presented in abstract form at AMERSA in Fall 2020. The content of this manuscript was not peer-reviewed nor published, other than a previous version of the abstract only for the conference. We have updated the manuscript to clarify this (Page 19, Prior Presentations).

Thank you. We are unable to share data because data are potentially identifying and contain sensitive patient information. These restrictions are imposed by our ethics committee. We have updated our cover letter to reflect this, and to provide the contact information for our ethics committee as requested. 

Thank you. We have updated our Supporting Information files per PLoS One guidelines.

5. Review Comments to the Author

Reviewer #1: This manuscript describes findings from a prospective qualitative study of 27 individuals with substance use disorder who were hospitalized during the COVID pandemic in an academic medical center in Portland, Oregon. What is challenging about the manuscript is that because only hospitalized patients with substance use disorder (PSUD) were interviewed, it is difficult to determine the extent to which the COVID-19 impacts describe in the paper are unique to this population or whether other patients in the hospital faced similar challenges. If this facility is a safety net hospital, many of the challenges around housing and other determinants of health in the context of the pandemic may not be all that different for other populations of patients. This limitation is not discussed in the paper.

Thank you for this important comment. Our team cares for patients hospitalized with substance use disorders. It is likely that other patients in the hospital who are marginalized, including patients who are homeless or unemployed but without substance use disorder, do face some similar challenges. One goal of this paper was to illuminate systems-level challenges for one group of highly marginalized patients. If care systems could reframe to address barriers to care for these patients, other patients may benefit as well. A key limitation is that we do not encompass challenges that other patients who experience marginalization in other ways, but who do not have substance use disorders, may face. We have updated to include this in our limitations (Page 16, paragraph 1), and it reads as follows:

“Fourth, it is possible that our research does not encompass all challenges faced by marginalized patients in healthcare systems. While we focus specifically on patients with substance use disorders and highlight that addressing challenges for these patients may benefit other marginalized groups, there may be other, unique, challenges that should be explored through further research with other marginalized patient populations.”

Introduction: The Introduction clearly and succinctly lays out the key issues that people with substance use disorders (PSUD) face in engaging with the healthcare system. The authors also note the rapid shifts in protocols within hospitals in response to the SARS-COV-2 pandemic. Attention to two issues in the Introduction would strengthen this section. First, the authors contend (page 3), “The SARS-COV-2 pandemic and mitigation strategies may have uniquely impacted people with acute illness and SUD.” This may indeed be the case but 1-2 more sentences about these unique impacts would orient the reader to the authors’ line of reasoning. At the same time, the authors cannot actually provide evidence to support this claim because the study only includes hospitalized PSUD. 

Thank you. We have updated them manuscript to read as follows (Page 3, paragraph 2):

“Negative and stigmatizing experiences during hospitalization can lead patients with SUDs to avoid presenting to the hospital until illnesses are dire, leave the hospital against medical advice, and trust healthcare providers less than patients without SUDs (15-18). Avoiding healthcare systems, or leaving the hospital without completing recommended therapy, carries increased risk during a pandemic (19). However, hospitals are also in a unique position to support patients with SUD, by providing evidence-based, compassionate care, including harm reduction tools, and by improving systems to meet the needs of people with SUD (20, 21).”

Second, the statement, “As we assess the ways in which hospitals’ responses to the pandemic have succeeded and failed, we must include voices of people at the margins, and specifically, voices of people with SUD,” should also be explained further. For readers who may be less well-versed in, for example, community-engaged research, further articulating what can be gained by examining the perspectives of PSUD in designing healthcare systems and processes would strengthen the rationale for the research.

Thank you for this comment. We have updated our introduction to include, after the sentence you’ve noted, the following:

“For example, by conducting qualitative research directly with people who are marginalized, we can understand how hospital policy changes have impacted them, which can help healthcare systems learn how to better care for patients in the future.” (Page 3 paragraph 1). 

Methods, Setting and Study Design: If the authors provided 1-2 more sentences about the broader hospital and the patients typically served, it would better contextualize the findings.

Thank you. We have updated the. First paragraph Setting and Study Design sections to read as follows:

 “We performed a prospective qualitative study at an urban academic medical center in Portland, Oregon. This hospital is a 576-bed Level 1 Trauma Center located on the top of a hill in the city [23]. OHSU sees a wide variety of patients for acute medical and surgical illness, and often receives transfers of patients from smaller hospitals across Oregon. In this study, all participants saw an inpatient addiction medicine consult service called the Improving Addiction Care Team (IMPACT). IMPACT is the only inpatient addiction medicine consult service in the state of Oregon, though some other hospitals provide medication for Opioid Use Disorder in inpatient settings. IMPACT cares for adults with SUD, including but not limited to opioids, alcohol, and methamphetamines. The team includes addiction medicine providers, social workers, and peers with lived experienced in recovery. Earlier work describes IMPACT development, intervention, and outcomes [24-27].”

In addition, it was somewhat difficult to tie together the state of SARS-COV-2 changes in the hospital during the specific timeframe of data collection. It appears that, to some degree, processes were stabilizing during the timeframe, but if the paragraph about SARS-COV-2 was woven into the description of when data were collected, it would be more clear.

Thank you for this helpful suggestion. We have reorganized our methods section to integrate the timeline of hospital-based changes and research (Page 4, paragraph 3). 

Also were the measures of accessing healthcare from an existing measure or developed specifically for this study?

We have updated the methods section to note the following:

“Participants first answered close-ended questions about demographics and experiences accessing healthcare. We used questions from the Brief Addiction Monitor [30] to capture recent substance use, patterns of use and recovery, and readiness to quit using substances. All other questions were designed by the research team specifically for this study.” (Page 5, paragraph 2)

Methods, Participants and Data Collection: Adding one sentence that describes the training of the interviewing team is also warranted. Also, consider moving the information about the sample size to this paragraph rather than including as the first sentence in the Results section.

Thank you. We have added the following to the methods (Page 5, paragraph 2): 

“The interview team consisted of three medical students, two female and one male, all interested in addiction medicine. Prior to the start of the study, two researchers who are experts in qualitative researcher (CN, HE) reviewed qualitative research steps for conducting interviews with the three medical students (CK, TV, DB), and oversaw the project for its duration. Then, one student with more qualitative experience (TV) conducted a mock interview with a student with less qualitative experience (DB) and gave feedback as to how to elicit rich qualitative data. Each of the three interviewers then did an initial one to two interviews. These interview recordings were transcribed, and the full research team reviewed transcripts and provided feedback on interviewing techniques.”

It is study team preference to include the sample size information in the Results section. If the reviewer feels it is essential to move this to the Methods, we are happy to do so. 

Results: The main findings are clearly conveyed, with themes supported by representative quotations from the interview transcripts. The findings about the challenges for individuals without cell phones are a major theme, but this reviewer wondered how individuals with phones perceived the shift to telehealth. In several sections of the Results, more explicit comparisons of themes for those with (51% of the sample) and those without phones (49%) would be beneficial, as it is difficult to gauge the perspectives of the half of the sample who did have phones.

Thank you. We have updated the text to include the following:

“While not having a phone was a unique stress for participants trying to access outpatient services, even participants with phones described challenges accessing care. Some participants who contacted their primary care office for a medical issue felt that in pre-COVID times, their primary care provider would have seen them in-office, but because of the pandemic and because they were very sick, they were sent to the emergency department instead.” (Page 11, paragraph 5)

Discussion: The Discussion does a nice job linking the findings to some of the literature on natural disasters and pandemics. Some of the tone about policy-makers is a bit beyond scope of the data (e.g., “Within Oregon, local, state, and national policy-makers implemented strategies meant to curb the spread of SARS-COV-2 for the masses, but instead inadvertently threatened the survival of people living at the margins”).

Thank you- we have removed this sentence. 

A second consideration is if or how themes perhaps shifted over the time period of data collection. Over the 6 weeks of data collection, it seems that the hospital restrictions loosened to some degree (based on the description in Methods about the impact of the pandemic within the hospital), but this is not discussed in terms of the findings.

Though hospital policy changed rapidly over the course of the study, most COVID-related changes (increased PPE, no hospital visitors) were consistent throughout the study period. There was some variability in which providers saw patients in person over the study period, however, this was also implemented differently by different hospital teams. Patients from the earliest enrollment date saw some clinicians in person, and patients near the end of the study had some virtual visits. We did not assess the impact of changing policy over time on study findings. We have updated our limitations to read as follows (Page 17, paragraph 1):

“Fifth, as is common in emergencies, policy within the hospital related to COVID-19 rapidly changed over the study period. We do not know to what degree these rapid changes may have impacted study results.”

The Limitations are a bit perfunctory regarding the limitation of focusing on an academic medical center. An additional sentence that elucidates why that is a limitation would be beneficial. Also, given a rapid escalation in cases this fall in Oregon, perhaps a sentence about how timing may affect the findings is warranted.

We have updated our limitations to read as follows:

“Our study has several limitations. First, we conducted our study at a single, academic medical center, which may limit generalizability, as community-based hospitals may see different challenges among patients with substance use disorders admitted to the hospital. Second, Oregon had relatively low rates of SARS-COV-2 infections during the study period, but this has since changed (31). It is possible that patient experiences and priorities differ depending on community prevalence of SARS-COV-2; however, the strains imposed by shuttered services, hospital policies, and high-risk transitions would likely only be further compounded in settings with high SARS-COV-2 cases (for example, in Philadelphia, where treatment clinics did close (48)). Third, our sample had low racial and ethnic diversity. Because Black, Indigenous, and People of Color disproportionately suffered from death and other complications of SARS-COV-2 (49) and also disproportionately experience opioid-related harms, including during the pandemic (50), future work should explore intersecting vulnerabilities of SUD, hospitalization, and acute crises like pandemics in this population. Fourth, it is possible that our research does not encompass all challenges faced by marginalized patients in healthcare systems. While we focus specifically on patients with substance use disorders and highlight that addressing challenges for these patients may benefit other marginalized groups, there may be other, unique, challenges that should be explored through further research with other marginalized patient populations. Fifth, as is common in emergencies, policy within the hospital related to COVID-19 rapidly changed over the study period. We do not know to what degree these rapid changes may have impacted study results. Sixth, additional research is needed to understand how the SARS-COV-2 pandemic directly impacted substance use access, practices, and use. While we found differences in use practices and access by primary substance, this question is better answered using quantitative data.” (Page 16, Paragraph 2)

The discussion about marginalized populations and telemedicine (page 15) would benefit from additional citations, as there have been concerns raised about unequal access to telemedicine both for SUD specifically but also for other health conditions.

We have updated our discussion section to include the following:

“In our study, nearly half of participants did not own a phone. Healthcare system shifts to telemedicine and virtual recovery supports were suddenly inaccessible to patients in need. Globally, shifts to telemedicine have impacted not only patients with substance use disorders, but also patients who are older, with disabilities, with lower socioeconomic status, and who are homeless [49].” (Page 17, paragraph 1)

References: The Reference list needs careful editing. There is variability in article titles in how the titles are capitalized, some author names seem erroneous (especially #24 and #37), there are inconsistencies in whether journal names are abbreviated or not, and some references pre-2020 lack volume and page numbers.

Thank you. We have updated our reference list throughout.

Reviewer #2: Thank you for inviting me to review this interesting and important paper. It presents the results from a qualitative study conducted with 27 patients hospitalized with substance use disorder during the initial phase of the COVID-19 pandemic in Spring 2020. The objective is clearly stated and appears to have been met (“evaluate how the SARSCOV-2 pandemic altered the experience of healthcare, acute illness, and care transitions among hospitalized patients with substance use disorder.”) The paper is well written and the findings are useful for informing improvements in clinical care for patients with SUD in pandemic settings and beyond. I especially appreciate the careful analysis, inclusion of patients’ voices, and discussion of important implications. I have one overall comment about the literature review and then some minor suggestions about ways that this paper could potentially be improved.

Thank you for reviewing our work!

The literature review feels a bit thin in places. Although COVID-19 is newly-emerged infectious disease, there may be experiences from other disasters that could be cited in the Introduction (some is cited in the Discussion section but that feels a bit late – this context would be helpful up front). There are also new studies and commentaries emerging that make me wonder if the lit review could be updated a bit? For example, PMC7419278 discusses some innovations implemented in one clinical setting during the pandemic with implications for care of patients with SUD. Similarly there are a few specific topics (e.g., stigma in the Introduction) where additional recent qualitative studies could be cited (PMID: 30884432). Taken together, these studies underscore the implications of the findings of the present study that if care is not improved for patients with SUD during pandemics (in part by reducing stigma and mistreatment), individuals may leave the hospital against medical advice, which has implications for their own health and others’ (esp. given the infectious nature of COVID). So strengthening the lit review throughout would help.

Thank you for your thoughtful review. We have updated in the second paragraph of the introduction to read as follows, and included both citations you identified (Page 3):

“Hospitalization may be a particularly challenging time for people with SUDs, as they commonly experience untreated withdrawal and pain and work to mitigate hospital-based social control (15). Negative and stigmatizing experiences during hospitalization can lead patients with SUDs to avoid presenting to the hospital until illnesses are dire, leave the hospital against medical advice, and trust healthcare providers less than patients without SUDs (15-18). Avoiding healthcare systems, or leaving the hospital without completing recommended therapy, carries increased risk during a pandemic (19). However, hospitals are also in a unique position to support patients with SUD, by providing evidence-based, compassionate care, including harm reduction tools, and by improving systems to meet the needs of people with SUD (20, 21).”

Minor comments:

Introduction:

Topics like “hospital-based social control” warrant some defining and explaining (as this, in particular, is a relevant issue for this population). Take care throughout to avoid the use of jargon or other terms that a general audience of this journal may not be familiar with.

Thank you. We have updated the introduction to note the following:

“Hospitalization may be a particularly challenging time for people with SUDs, as they commonly experience untreated withdrawal and pain and work to mitigate hospital-based social control (e.g. the rules and regulations that dictate acceptable behavior in hospitals).”

Methods:

It is unclear what a “prospective qualitative study” is. (also in the Abstract). If participants were enrolled sequentially that is fine but does make this “prospective” in the sense that it was a cohort or other type of study looking forward. And it does not sound like patients were interviewed more than once.

We agree- the word prospective is confusing. We have removed this term. (Page 4, paragraph 2).

Please explain why there weren’t more efforts to implement telemedicine for this population initially. Experiences from other settings have shown that hospitals can provide wifi enabled devices to inpatients for consults (and even extending to after discharge).

Thank you. We have updated our manuscript to include the following (Page 4, paragraph 3):

“IMPACT leadership tried to secure personal electronic devices (e.g. smartphones, tablets) for patients to support video communication, but issues of cost and competing priorities were barriers and efforts did not come to fruition until after the study window.”

It sounds like data collection was collected remotely but that is not specifically explained. Nor are the 

recruitment or consent/enrollment processes if this was all remote/virtual. Please explain these procedures and comment on any related challenges or facilitators as this could be informative to the research community. Similarly, please include more details on IRB review and consent given that there was an unprecedented shift in research operations and oversight during this time period.

We have updated our methods to include the following:

“Our institutional IRB waived the requirement for written consent given the inability to meet patients in person because of university-wide restrictions, and because of the minimal risk associated with our study. To recruit patients, we called the rooms of hospitalized patients with SUDs who were referred to IMPACT. We ascertained patient interest in learning about the study, and verbally consented patients to participate. Patients were offered $25 gift cards to thank them for participating.” (Page 5, paragraph 2)

What does “at a semantic level” mean? This should be defined in the text.

We have updated the text to include the following: 

“We conducted a thematic analysis, using an inductive approach, at a semantic level (e.g. where interview data were taken at face value) (33, 34).” (Page 6, paragraph 4)

Results: what were the reasons for declining participation or ineligibility?

We have updated the text to include the following:

“We approached 42 patients via telephone to participate, and enrolled 27 participants. Of the 15 patients who declined to enroll, 8 said that they were not interested, 2 were managing acute illness and did not want to be on the phone, and 5 did not provide a reason for not participating.” (Page 6, paragraph. 5)

Discussion: see earlier comment about moving some of the literature on disasters to the Introduction. This is optional but I feel would help to set the stage earlier on.

Thank you. In early drafts of this manuscript, the introduction included literature on disasters. After iterative review from our research team, we felt that this was out of place in the introduction, and moved this information to the discussion section. Our sense is that reorganizing to include in the intro again may still feel out of place. However, if the reviewer feels strongly we should reorganize to describe the disaster literature in the introduction, we are happy to do so. 

Limitations: there is increasing evidence that Black and Hispanic individuals are at higher risk of experiencing opioid-related harms, not just COVID.

Thank you for this important comment. In the limitations, we have updated to include the following:

“Because Black, Indigenous, and People of Color disproportionately suffered from death and other complications of SARS-COV-2 (49) and also disproportionately experience opioid-related harms, including during the pandemic (50), future work should explore intersecting vulnerabilities of SUD, hospitalization, and acute crises like pandemics in this population.” (Page 17, paragraph 1)

---

## [Editor Report · Decision Letter 1]

17 Feb 2021

Understanding the impact of the SARS-COV-2 pandemic on hospitalized patients with substance use disorder

PONE-D-20-30950R1

Dear Dr. King,

We’re pleased to inform you that your manuscript has been judged scientifically suitable for publication and will be formally accepted for publication once it meets all outstanding technical requirements.

Kind regards,

Cynthia Sieck

Academic Editor

PLOS ONE
---

## [Editor Report · Acceptance letter]

19 Feb 2021

PONE-D-20-30950R1 

Understanding the impact of the SARS-COV-2 pandemic on hospitalized patients with substance use disorder 

Dear Dr. King:

I'm pleased to inform you that your manuscript has been deemed suitable for publication in PLOS ONE. Congratulations! Your manuscript is now with our production department. 

Kind regards, 

on behalf of

Dr. Cynthia Sieck 

Academic Editor

PLOS ONE